# Genetic analysis and fine mapping of a qualitative trait locus *wpb1* for albino panicle branches in rice

**Zhongquan Cai**[1,2☯]**, Peilong Jia**[2☯]**, Jiaqiang Zhang**[3]**, Ping Gan**[1]**, Qi Shao**[1]**, Gang Jin**[4]**, Liping Wang**[4]**, Jian Jin**[1]***, Jiangyi Yang**[1]***, Jijing Luo**[1]***

**1** College of Life Science and Technology, State Key Laboratory for Conservation and Utilization of Subtropical Agro-Bioresources, Guangxi University, Nanning, China, **2** Institute for New Rural Development, National Demonstration Center for Experimental Plant Science Education, College of Agriculture, Guangxi University, Nanning, Guangxi, China, **3** Research and Development Centre of Flower, Zhejiang Academy of Agricultural Sciences, Hangzhou, China, **4** Guangxi Subtropical Crops Research Institute, Nanning, China

☯ These authors contributed equally to this work.

* jjluo@gxu.edu.cn (JL); yangjy598@163.com (JY); jinjianes@outlook.com (JJ)

**Data Availability Statement:** All relevant data are within the manuscript and its Supporting Information files.

**Funding:** Funded by JL, 31671646, The National Natural Science Foundation of China (CN) JL,

## Abstract

Chloroplast plays an important role in the plant life cycle. However, the details of its development remain elusive in rice. In this study, we report the fine-mapping of a novel rice gene *wpb1* (white panicle branch 1), which affects chloroplast biogenesis, from a tropical *japonica* variety that results in an albino panicle branches at and after the heading stage. The *wpb1* variety was crossed with Nipponbare to generate the $F_2$ and $BC_1F_2$ populations. Green and white panicle branch phenotypes with a 3:1 segregation ratio was observed in the $F_2$ population. Bulked segregant analysis (BSA) based on whole genome resequencing was conducted to determine the *wpb1* locus. A candidate interval spanning from 11.35 to 23.79M (physical position) on chromosome 1 was identified. The results of BSA analysis were verified by a 40K rice SNP-array using the $BC_1F_2$ population. A large-scale $F_2$ population was used to pinpoint *wpb1*, and the locus was further narrowed down to a 95-kb interval. Furthermore, our results showed that the expression levels of the majority of the genes involved in Chl biosynthesis, photosynthesis and chloroplast development were remarkably affected in *wpb1* variety and in $F_2$ plants with a white panicle branch phenotype. In line with the results mentioned above, anatomical structural examination and chlorophyll (Chl) content measurement suggested that *wpb1* might play an important role in the regulation of chloroplast development in rice. Further cloning and functional characterization of the *wpb1* gene will shed light on the molecular mechanism underlying chloroplast development in rice.

## Introduction

Chlorophyll (Chl) is green pigments found in cyanobacteria and the chloroplasts of algae and green plants on Earth. Chlorophyll is essential, allowing plants to absorb light energy and perform photosynthesis [1]. In addition, chlorophyll plays an critical role in human health, including cancer treatment and prevention [2]. Chlorophyll molecules are structural

2015, Guangxi Hundred-Talents Program ZC, 2014GXNSFBA118126, 2018GXNSFBA281109, 2018GXNSFBA138022, The Natural Science Foundation of Guangxi, China JY, 2016, The Project of High Level Innovation Team and Outstanding Scholar in Guangxi Colleges and Universities (Third batch) JZ, 2016. The study abroad program of the Zhejiang Academy of Agricultural Sciences. The funders had no role in study design, data collection and analysis, decision to publish, or preparation of the manuscript.

**Competing interests:** The authors have declared that no competing interests exist.

constituents of photosystems that are embedded in the thylakoid membranes of chloroplasts. Thus, chloroplasts are important organelles for the fixation of light energy for the life of plants, and any other life in our biosphere. The initiation and development of chloroplasts are jointly regulated by the nuclear genome and chloroplast genome [3]. Previous studies have reported that mutation of genes that control chloroplast development or chlorophyll synthesis leads to albinism in some specific parts of the plant [4–7]. The mutations seriously affect the photosynthetic efficiency, resulting in reduced production and even causing the developmental arrest of plants. Many related genes have been cloned and functionally characterized. In *Arabidopsis*, *ALB3*, which is encoded by a nucleic gene, is transported into chloroplasts and embedded into the chloroplastic membrane, where it is involved in the composition of the chloroplast enzymatic complex. Loss of function of *Alb3* gives rise to a white or yellow cotyledons and leaves, leading to stop growing beyond the seedling stage [8]. An *ylc1* mutant of *indica* rice 9B, which was induced by $^{60}$Co radiation, resulted in a decrease in chlorophyll and lutein content at different temperatures [9]. In tobacco, an *ali* albino mutant with an albino phenotype was obtained by irradiating BY-4 tobacco with a 14N ion beam, and the expressions of plastid-encoded genes *rbcL* and *psbA* in the mutant was down-regulated dramatically [10]. *FA85* is a natural mutant of winter wheat. The leaf colour of the mutant is whitened, and the abundances of ATPase-γ and GP1-α are up-regulated, while the biosynthesis of other chloroplastic proteins is remarkably inhibited [11]. Moreover, in maize, the *ppr4* gene encodes a chloroplast-targeted protein containing a *PPR* sequence and RNA recognition elements. PPR4 directly targets the intron of *rps12* and affects the accumulation of *rps12* mRNA in chloroplasts [12].

The molecular mechanism of rice albino phenotypes is complex. At present, more than 100 genes related to leaf colour alternation in rice have been reported (http://www.gramene.org/, up to now). These genes are involved in chloroplast development regulation, chlorophyll synthesis or degradation, etc. [13]. For example, rice *v3* and *stl* genes impair the synthesis of plastid DNA and hinder chloroplast differentiation [14]; The *virescent-l* (*v1*), *virescent-2* (*v2*), and *vyl* genes were found to hinder the formation of chloroplasts, resulting in white streaks on rice leaves. In their WT plants, these genes highly express in the second stage of chloroplast development [4, 6, 15–17]; Rice *wsl* and *wsl4* genes, which encoded proteins WSL and WSL4, respectively, involved in RNA metabolism in chloroplasts. Their mutations caused abnormal splicing of *rpl2*, *ndhA*, *atpF* and other chloroplast gene transcripts, resulting in leaf streaks [18, 19]. In addition, another class of genes, such as *WSL3* and *YSS1*, encode the proteins that targeted the chloroplast nucleoid and played a key role in regulating the expression of genes related to Plastid-encoded plastid RNA polymerase (PEP). Their mutations resulted in a white streak phenotype in rice [20, 21].

At present, the albino phenotypes have been observed on the panicle, seedlings and leaves of rice, for example, in addition to two albino mutants *v1* and *v2* mentioned above [6, 22], another two classical panicle colour mutants *wp1* and *wp2*, which were used as morphological markers in the field, show albino coloration on the whole panicles. The *wp1* locus was located on chromosome 7 [22, 23]. Li *et al*. fine mapped a white panicle controlling gene *wp(t)* between the markers SSR101 and SSR63.9 on chromosome 1, which was close to the morphological marker *wp2* and might be another allelic mutation of *wp2* [24]. The *wp1* and *wp2* mutants were all whitened along the entire panicle. In addition, the mutations of other genes that have been reported to control the related traits of white panicle in rice, including *st-wp* [24], *slwp* [25], *alsm6* [26], *st-fon* [27], *wp4* [28], *wlp1* [29], *wlp6* [13], *wslwp* [30], and *wsp1* [31]. However, with the exception of the albino phenomenon observed on the whole panicle, including the rachises, branches, and glumes, it is rare to see the albino phenotype observed only on panicle branches.

SNPs (single nucleotide polymorphisms) and InDels (insertions-deletions) are the most common genetic markers in the genome. They have the characteristics of large numbers and rich polymorphisms and are often used in QTL analysis. Bulked segregant analysis (BSA) is a rapid method for locating target trait genes and it was initially used in lettuce [32]. This method constructs two mixing pools by mixing individuals with extreme traits in the segregating population [33]. By analysing the differences between SNPs and InDels between the two mixing pools, we can quickly locate the molecular markers closely linked with the target gene. This method has been widely used in gene mapping of *Arabidopsis thaliana* [34], rice [35, 36] and maize [37], etc.

In this study, a rice white panicle branch variety *wpb1* was characterized. Its panicle branches showed an albino phenotype with stable heredity. The branches of young panicles were observed to be whitened from the heading stage, and the whitened colour was kept even until the ripening stage. However, the rachis and glumes of the panicles of *wpb1* show normal or light green colour. Therefore, the variety does not show a defect in flowering and seed setting. To our knowledge, the panicle branch albino mutant material is valuable for frescamente ornament, due to its snow-white panicle and streaked leaves, especially in autumn. The locus underlying the *wpb1* phenotype was preliminarily mapped by BSA and 40K rice SNP-array analysis, and then the locus was fine-mapped to the 95-kb interval on chromosome 1. Our results laid a foundation for further cloning of the gene and its functional characterization. Likewise, the study also shed light on the physiological process of chlorophyll biosynthesis, the development of chloroplasts, and photosynthesis.

## Materials and methods

### Plant materials

The inbred rice variety *wpb1* (*Oryza sativa* L. *subsp. japonica*), with a white panicle branch, derived from natural mutation. The field trials were performed in an experimental field at the campus of Guangxi University, Nanning City, China (E108˚22′, N22˚4). The cross between *wpb1* and NIP (Nipponbare) (*Oryza sativa* L. *ssp. japonica*) was used to generate population materials to map the locus underlying the *wpb1* phenotype. The *wpb1* variety was selected as the female parent, and NIP was selected as the pollen donor. Then, the $F_2$ population was generated via self-crossing of $F_1$.

### Genetic analysis of the white panicle branch phenotype

The colour of the panicle branch at the heading stage of all plants was investigated under field trial conditions. The number of white panicle branch plants and green panicle branch plants for every population, including the parents, $F_1$, $F_2$, $BC_1F_1$, and $BC_1F_2$, were recorded. The segregation ratios in the $F_2$ and $BC_1F_2$ populations were analysed, and the genetic model was inferred and tested using a Chi-square test with SPSS software.

### High-throughput sequencing

The genomic DNA of young and healthy leaves was extracted using the DNeasy 96 Plant Kit (Qiagen, Valencia, CA). The integrity of each DNA sample was examined by 1% agarose gel electrophoresis. DNA purity was determined using a NanoPhotometer® spectrophotometer (IMPLEN, CA, USA). DNA concentration was measured using a Qubit® DNA Assay Kit in a Qubit® 2.0 Flurometer (Life Technologies, CA, USA). Equal amounts of DNA (1.5 μg/sample) from 30 $F_2$ plants with white panicle branches were mixed to form the white panicle branch bulk sample (W-pool), and those from another 30 plants with normal (green) branches

were mixed to form the normal branch bulk sample (G-pool). Sequencing libraries were constructed using a Truseq Nano DNA HT sample preparation kit (Illumina USA) following the manufacturer's instructions. In brief, the DNA sample was sheared to a 350-bp fragment by sonication. The obtained DNA fragments were end-polished, A-tailed, and ligated with a full-length adapter for deep sequencing. The libraries were sequenced using an Illumina HiSeq 4000 platform with a 20× depth. Sequence data were analysed by Novogene (Beijing, China). To ensure that reads were reliable and without artificial bias, quality control (QC) procedures were set as follows: reads with ≥10% unidentified nucleotides were removed, reads with > 50% bases having phred quality < 5 were removed, reads with > 10 nt aligned to the adapter were removed, ≤10% mismatches were allowed, putative PCR duplicates generated by PCR amplification in the library construction process were removed. The genomic information of NIP was obtained from NCBI. The genomic DNA data of NIP was downloaded from the website: ftp://ftp.ensemblgenomes.org/pub/plants/release-36/fasta/oryza_sativa/dna/. Mapping to the reference genome BWA (Burrows-Wheeler Aligner) was used to align the clean reads of each sample against the reference genome (settings: mem -t 4 -k 32 -M -R) [38].

## Bulked segregant analysis (BSA)

SNP/InDel detection and annotation variant calling were performed for all samples using the Unified Genotyper function in GATK software [39]. SNP was used as the variant filtration parameter in GATK (settings: filterExpression QD < 4.0 || FS > 60.0 || MQ < 40.0, G_filter GQ<20, cluster WindowSize 4). InDel was filtered by the variant filtration parameter (settings: filter Expression QD < 4.0 || FS > 200.0 ||Read PosRankSum < -20.0 || Inbreeding Coeff < -0.8). ANNOVAR [40] was used to annotate SNPs or InDels for the reference genome. The homozygous SNPs/InDels between two parents were extracted. The read depth information for homozygous SNPs/InDels in the offspring pools was obtained to calculate the SNP/InDel index (frequency) [33]. The genotype of one parent was used as the reference statistic for the the SNP/InDel index. The points at which the SNP/InDel index in both pools was less than 0.3 were filtered out. Sliding window methods were used to present the SNP/InDel index of the whole genome. The average SNP/InDel index in each window was used as the SNP/InDel index for this window. A window size of 1 Mb and a step size of 10 Kb were used as default settings. The difference in the SNP/InDel index of the two pools was calculated as the delta SNP/InDel index. The intervals of the delta all index in the 95% confidence interval of the permutation test were selected as candidate loci. Genes with SNPs causing stop gain or loss, that were non-synonymous and spliced, or those with InDels causing stop gain or loss, or frame shift mutations in their corresponding alleles were selected as the candidate genes in the 95% confidence interval.

## Validation of BSA results by 40k rice SNP-array analysis

The result of the BSA was validated by 40k rice SNP-array, a whole-genome single nucleotide polymorphism (SNP) array with 40k SNP and InDel markers, using the $BC_1F_2$ population. The total genomic DNA from young and healthy leaves of $BC_1F_2$ population plants was extracted using the DNeasy 96 Plant Kit (Qiagen, Valencia, CA). Equal amounts of DNA from 30 $BC_1F_2$ plants exhibiting white panicle branches were mixed to form the white branch bulk sample (W-pool), and those from another 30 plants with normal green panicle branches were mixed to form the normal branch bulk sample (G-pool). DNA amplification, fragmentation, chip hybridization, single base extension, staining and scanning were conducted by the greenfafa Science and Technology Research Institute Co., Ltd. (Wuhan, China) following the Infinium HD Assay Ultra Protocol (http://www.illumina.com/).

## Fine mapping of *wpb1*

The candidate locus of *wpb1* was fine-mapped with high-resolution linkage analysis by map-based cloning. For the analysis, DNA was isolated from the two parental lines and the $F_2$ population. The informative molecular markers were used for genotyping each plant of the $F_2$ population, various recombinants in the target region were identified, and the linkage relationship between markers and the *wpb1* locus was analysed for gene mapping. The flanking SSR markers were obtained from Gramene (http://www.gramene.org/). The genomic DNA sequences of the two parents, *wpb1* and NIP, obtained from Illumina sequencing were used to develop InDel markers for fine mapping (S1 Table).

## Analysis with quantitative real-time PCR (qPCR)

To examine the expression differences of the genes involved in Chl biosynthesis, photosynthesis, and chloroplast development in *wpb1* and NIP, RNA was extracted from young panicle branches of *wpb1* and NIP at stage In8 using an RNA Prep Pure Plant Kit (Tiangen Co., Beijing, China). Total RNA was reverse transcribed using a FastKing kit KR123 (Tiangen Co., Beijing, China). For transcriptional analysis of Chl biosynthesis-associated, chloroplast development-associated, and photosynthesis-associated genes (*HEMA1*, *CAO1*, *PORA*, *V1*, *v2*, *rpoA*, *rpoB*, *Cab1R*, *Cab2R*, *psaA*, *psbA*) in rice (S2 Table) [4, 6, 41–44], qPCR analyses were performed using a SYBR Premix Ex TaqTM kit (Takara) on a LightCycler 480 II Real-Time PCR System (Roche). The relative quantification of gene expression data was performed as described in Livak & Schmittgen [45]. Actin coding gene, *actin-7* (Os11 g0163100), was used as an internal reference. The specific primers for qPCR were designed according to Zhang et al. [44] (S2 Table).

## Chlorophyll content measurement

The leaf chlorophyll content was determined according to the method described by Qiu *et al*. [46]. The plant materials were cut into 1 mm small pieces. Next, 50–100 mg of the samples were placed into a 10 mL graduated test tube with a stopper. Then, 2 mL DMSO was added into the test tube, and the samples were immersed in DMSO. The tubes were incubated in a 65˚C incubator in the dark until all the samples turned white or transparent. The chlorophyll in the leaf samples was dissolved into DMSO. While the solutions cooled down, 8 mL 80% (v/v) acetone was added and mixed well. Then, the absorbance of the solution was determined at 663.6 and 646.6 nm by spectrophotometry. Chlorophyll concentration was calculated with the following formulas: $Chl_a$ $(mg·L^{-1})$ = 12.27×A663.6–2.52×A646.6; $Chl_b$ $(mg·L^{-1})$ = 20.10×A646.6–4.92×A663.6; $Chl_T$ $(mg·L^{-1})$ = $Chl_a$+$Chl_b$ = 7.35×A663.6+17.58×A646.6.

## Chlorophyll fluorescence analysis

The stalks, branches, and glumes of the spikelet from *wpb1*, NIP and $F_2$ plants were used for chlorophyll fluorescence analysis after heading. A Research Stereomicroscope System (Olympus SZX-16 Stereo Microscope, Olympus Corporation) was used to capture chlorophyll fluorescence images following the manufacturer's protocol. The microscopy images were photographed with the following settings: exposure time 300 ms, ISO 200 under bright field, exposure time 1 s, ISO 1600 under RFP.

## Histological analysis

For histological analysis, fresh panicle branch samples from *wpb1*, NIP and $F_2$ plants were cut into ultrathin transverse sections by the Leica CM1860 UV Cryostat (Leica Biosystems

Nussloch GmbH, Germany), following the manufacturer's protocol. Microscopy was performed on a Leica DM4 B upright digital research microscope.

For TEM analysis, panicle branch samples of *wpb1* and NIP were soaked in primary fixation buffer (2.5% glutaraldehyde) and post-fixed for 2 h in secondary fixation buffer (1% OsO4 in 100 mM cacodylate buffer, pH 7.4). The fixed samples were dehydrated in an ethanol series, embedded in resin, and stained by uranyl acetate together with lead citrate for 15mins, separately. Using Reichert-Jung ULTRACUT E ultra-thin slicing machine (Austria), 70 nm slicing, copper mesh fishing. Ultra-thin sections were observed by TEM (JOEL JEM-1200 electron microscope).

## Results

### The white panicle branches were observed in *wpb1* variety

A tropical *japonica* rice variety *wpb1* (*Oryza sativa* L. *subsp. japonica*) has normal plant architecture and normal colour of the aboveground parts at the vegetative stage in summer, similar to the majority of extant rice varieties. However, at the heading stage, the *wpb1* variety exhibits white panicle branches including the primary and secondary branches, as well as the white lemma and palea (Fig 1; S1A Fig). Although the glume and rachises were coloured slightly after heading, the albino phenotype of branches was observed with no significant alteration from heading to ripening stage (Fig 1B–1E; S1A Fig). The colour of the albino glumes turned light green at the heading stage and then turned purple later. In particular, the *wpb1* showed even more severe defect under low temperature conditions (15–25°C); for example, the branches turned purple after heading in the late autumn in Nanning (S2C Fig). In addition, white streaks appeared in some young leaves of *wpb1* at the tillering stage under low temperature conditions (S2A and S2B Fig).

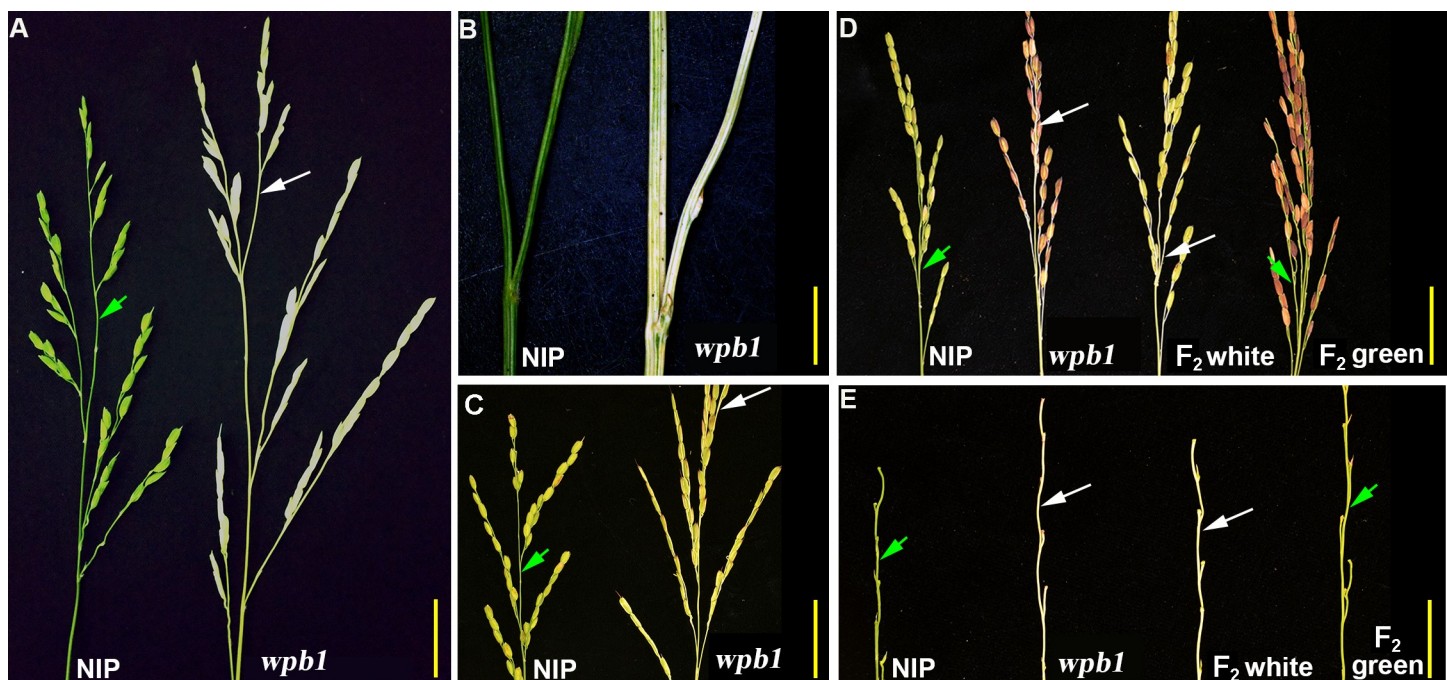

**Fig 1. Characterization of the phenotypes of *wpb1* and F$_2$ populations.** A-C. The panicles of NIP and *wpb1* mutant. Young panicles examined 3-day before heading (A). Magnification of the panicle branch (B). Mature panicle branch (C). D-E. The panicles of parents and the individuals of F$_2$ populations. Mature panicles (D). Mature branches (E). Bar, 3cm (A). 0.5cm (B). 5cm (C-D). 1cm (E).

## Genetic analysis of the white branch phenotype

To understand the genetic basis of the white panicle branch phenotype in the *wpb1* variety, NIP was crossed with *wpb1* to generate $F_1$ and $F_2$ populations. The primary and secondary branches of $F_1$ exhibited green colour that was comparable to that of NIP (S1B Fig). In the $F_2$ population, green/white panicle branch segregation was observed from the phenotypic investigations during two rice seasons (S1C and S1D Fig). In 2017, among the 365 individuals of the $F_2$ population, 90 exhibited white branches, and 275 showed green branches. In the $F_2$ population of 2018, 63 showed white branches, and 196 had green branches. The chi-square test showed that the segregation ratio of green: white $\approx$ 3:1 (Table 1).

To further confirm the heredity of white panicle branches in *wpb1*, the $F_2$ individual plants with white streaked leaves, green glumes, and white branches were backcrossed to NIP to obtain a $BC_1F_2$ line. In line with the observations in the $F_1$ and $F_2$ populations, the branch colour of $BC_1F_1$ was green, and without the leaf white streaks. The colour phenotype of the $BC_1F_2$ population was also segregated, similar to that in the $F_2$ population. Among a total of 206 plants, 151 showed green branches, and 55 had white branches. The chi-square test showed that the segregation ratio of the $BC_1F_2$ population was the same as that observed in the $F_2$ population (Table 1). Taken together, the results indicated that the white panicle branch in *wpb1* is controlled by a single recessive nucleic gene. Moreover, the phenotypes of white-streaked leaves were observed co-segregating with those of white panicle branches. In the $F_2$ population, all the plants with white-streaked leaves had white panicle branches, and only some of the plants with white panicle branches had white streaked leaves. In the $BC_1F_2$ population (206 plants), white streaked leaves and white panicle branches were completely co-segregated.

## The candidate loci of *wpb1* were located on chromosome 1

Bulked segregant analysis coupled to whole genome sequencing is an efficient and rapid way to target the candidate loci for qualitative traits [32, 37]. In this study, genomic DNA of the G-pool (green branch pedicel) and W-pool (white branch pedicel) of the $F_2$ population and a parent (*wpb1*) were sequenced to perform BSA analysis. A total of 32.443G raw data were obtained from the sequencing. Raw data were filtered to remove low quality data, and 32.402G of clean data was obtained for further analysis. The raw data of each sample ranged from 6055.43 to 15501.463 M. The average Q20 and Q30 were 96.5% and 94.61%, respectively. The GC content ranged from 44.61% to 46.53%. The percentage ranged from 97.97% to 98.2% of all samples mapped to the reference genome. The average coverage depth of the reference genome (excluding the N region) ranged from 12.84× to 34.14×. Subsequently, SNP and InDel calling was performed to identify SNP and InDel genotyping, and 1,096,326 SNP and 205,407 InDel polymorphic markers were obtained. Among them, 744,778 SNPs and 181,731 InDels were selected as informative polymorphic markers (frequency > 0.3, depth > 7, and both parents were present) to calculate the index and delta index. Two candidate intervals were located on chromosomes 1 and 3 by permutation test with 1000 permutations per test (Fig

**Table 1. Genetic analysis of white branch mutant.**

| Population | Year | Total plants | Green Branch | White Branch | Ratio | $\chi^{2a}$ |
|---|---|---|---|---|---|---|
| Nip×*wpb1* $F_2$ | 2017 | 365 | 275 | 90 | 3:1 | 0.0115 |
| Nip×*wpb1* $F_2$ | 2018 | 259 | 196 | 63 | 3:1 | 0.0318 |
| Nip//Nip/*wpb1* $BC_1F_2$ | 2018 | 206 | 151 | 55 | 3:1 | 0.1551 |

a$\chi^2 < \chi^2_{0.05} = 3.84$ is considered a significant difference at the $P<0.05$ level.

2C). Twenty-seven SNPs and 85 InDels were selected as the candidate loci, whose indexes were close to 0.1 in the W-pool and close to 0.9 in the G-pool (Fig 2A and 2B). Eleven candidate genes that were altered by causing missense, premature termination, losing termination codon mutations or variable splicing sites were selected according to ANNOVAR annotations. Among them, 10 were located on chromosome 1 and 1 was located on chromosome 3 (S3 Table).

To validate the BSA result obtained from Illumina sequencing, a genetic background analysis was carried out using a 40k rice SNP array. Thirty green and 30 white panicle branched plants were selected from the $BC_1F_2$ population to construct the mixing pool. The result indicated that the only candidate interval was mapped to physical position from the 9.54 to 14.28 M region of chromosome 1, and the majority of the interval overlapped with the candidate interval mapped in the above BSA analysis (Fig 2D). Furthermore, the candidate mapped to chromosome 3 in Illumina BSA analysis was excluded. The interval mapped on chromosome 1 is the most likely candidate for the *wpb1* locus.

## Fine mapping of *wpb1*

To further fine map *wpb1*, a large-scale $F_2$ population (2867 plants) was generated to narrow down the locus into a small region. Meanwhile, new molecular markers were developed in the preliminarily mapped interval by BSA analysis based on the sequencing data of NIP and the *wpb1* variety. Genotyping was conducted using newly developed PCR markers (S1 Table). The linkage analysis of the phenotype was performed with marker genotypes. Four key informative recombinants were identified to narrow down the mapping region. Finally, the *wpb1* locus was narrowed down to a 95-kb genomic interval with flanking markers M8 and M10, and this region contains 17 annotated genes (Fig 3, S4 Table).

## The expressions of genes associated with Chl biosynthesis, photosynthesis, and chloroplast development were altered in *wpb1*

In general, the branches and the spikelets of the panicle turn green before heading with the maturation of chloroplasts. The albino phenotype of the *wpb1* variety suggests that the processes of the chloroplasts development and/or Chl biosynthesis were impaired in its panicle branches, and therefore resulting in the inhibition of photosynthesis in panicle branches of the mutant. To validate this possibility, we next examined whether the related processes were compromised in *wpb1* by comparing the relative expression levels of the genes involved in Chl biosynthesis, chloroplast development, and photosynthesis of *wpb1* with those in WT (Nipponbare) and $F_2$ plants. The expression levels of Chl biosynthesis-related genes [44], such as *CAO1* (CHLOROPHYLLIDE A OXYGENASE1), *PORA* (encoding NADPH-dependent protochlorophyllide oxidoreductase), and *HEMA1* (encoding glutamyl tRNA reductase), was significantly downregulated in *wpb1* in contrast to NIP. Similar regulation patterns were observed in the white panicle $F_2$ plants. Furthermore, the expression of these genes in green branched plants was significantly higher than in the white branched $F_2$ plant, which was consistent with the difference between their two parents (Fig 4A).

For the chloroplast development-associated genes, we investigated both nuclear-encoded genes *V1* [4], *V2* (both encoding plastidal guanylate kinase) [6] and plastid genome-encoded genes *rpoA* and *rpoB* (encoding the PEP core α, and β subunit, respectively) [42]. *rpoB* was significantly upregulated in the white panicle branch in *wpb1* relative to the green panicle branch in NIP. Moreover, in line with this result, the expression of the gene in the white branched $F_2$ plant was also significantly upregulated. It is worthy to note that *rpoA*, *V1* and *V2* were all

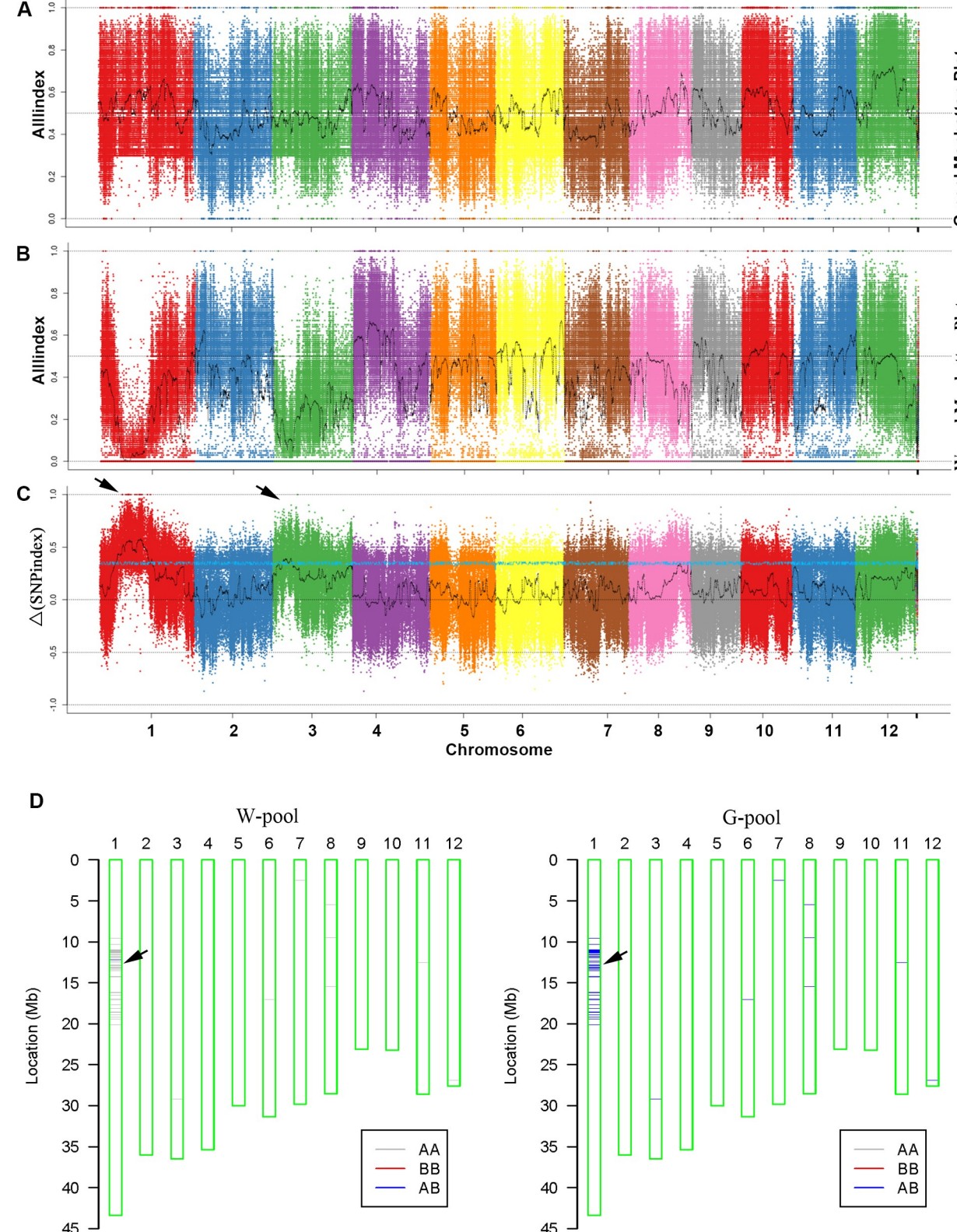

**Fig 2. The preliminary mapping of *wpb1*.** A-C. Manhattan plot of the all (SNP and InDel) index and delta index. All index graphs of G-pool (A), W-pool (B) and the all index (C) from BSA-seq analysis. The X-axis represents the position of 12 chromosomes, and the Y-axis represents the All index. The all index was calculated based on a 1 Mb interval with a 1 kb sliding window. The delta all index graph (C) was plotted with a

permutation test with 1000 permutations per test ($P < 0.05$). The blue line represents the confidence interval (C). D. Comparative genetic background analysis of the W-pool and G-pool of $BC_1F_2$ detected by 40k SNP array. The green square bar indicates the chromosomes. The grey lines indicate the SNP loci with homozygous genotypes AA (allele for white branch pedicel), red lines indicate the other homozygous genotypes BB, and the blue lines indicate the heterozygous genotypes AB. The candidate interval is on chromosome 1.

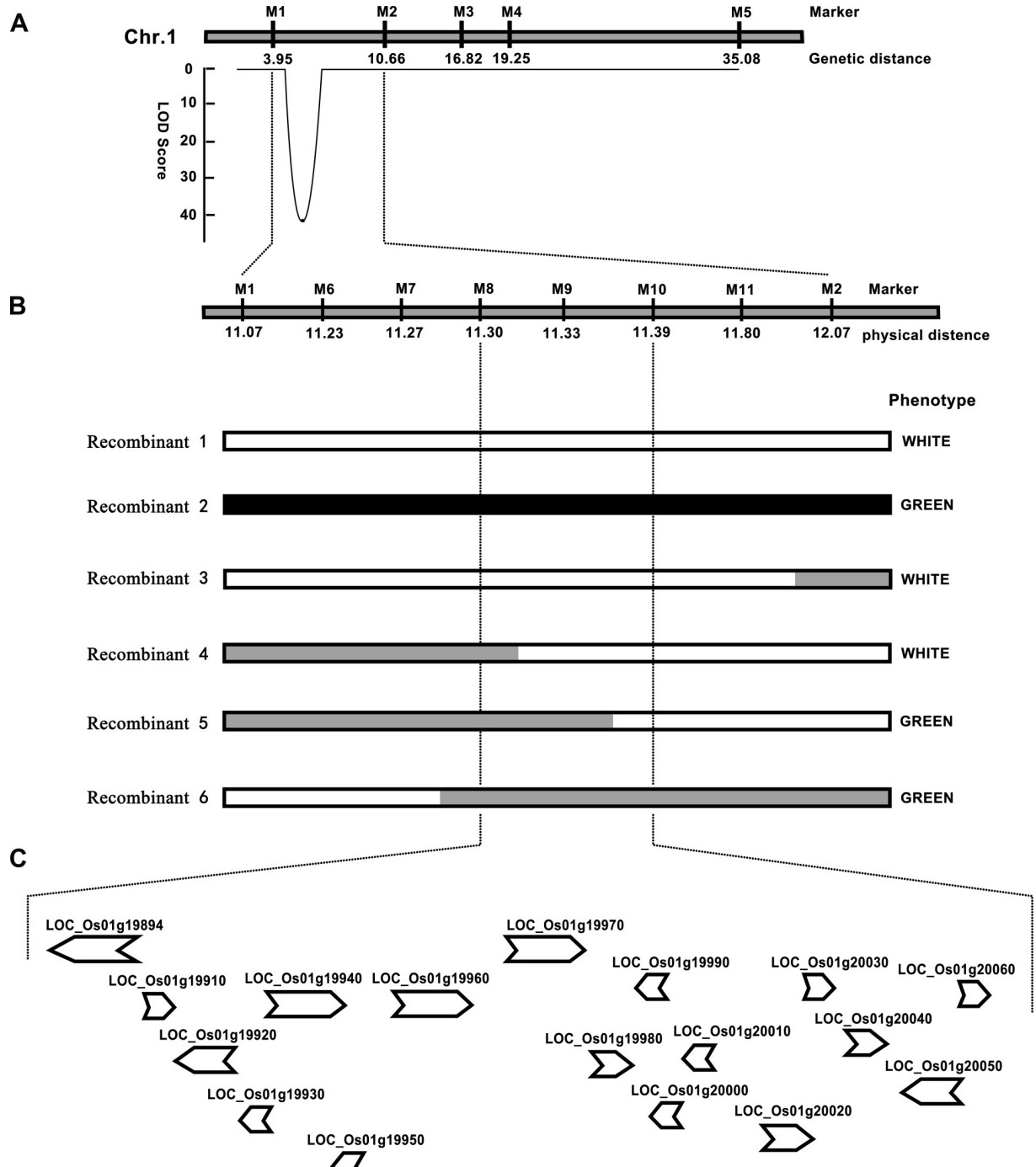

**Fig 3. Fine mapping of *wpb1*.** A. Preliminary mapping of *wpb1*. B. Physical map of the *wpb1* locus. Two key recombinants delimited the mapping region. C. Putative ORFs in the mapping region.

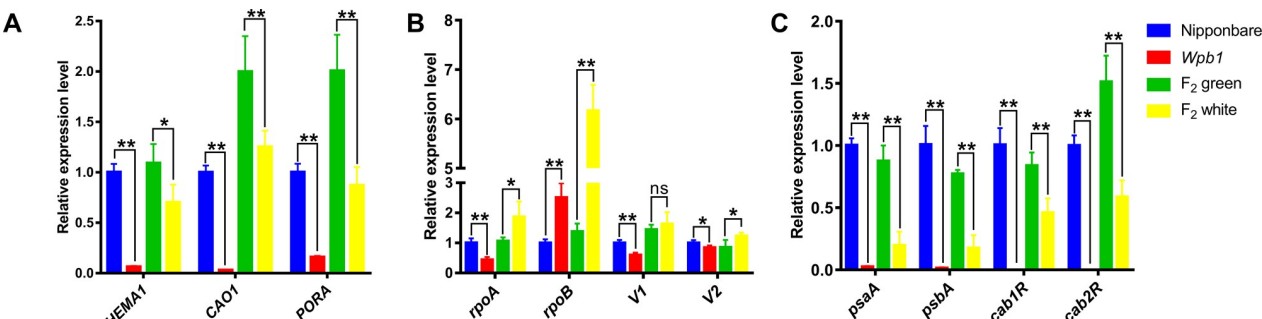

**Fig 4. qRT-PCR analysis of genes associated with chloroplast biogenesis, chloroplast development, and photosynthesis in parents and F₂ populations.** A. Chloroplast biogenesis-associated genes. B. Genes associated with chloroplast development. C. Photosynthesis-associated genes. The relative expression level of each gene in the mutant and F₂ plants was analysed with qRT-PCR and normalised using the actin gene as an internal control, and its expression level in NIP plants was set to 1.0. * (*P*<0.05), ** (*P*<0.001).

downregulated in *wpb1*, whereas, in contrast, they were upregulated in the white branched F₂ plant (Fig 4B).

In the case of the photosynthesis-associated plastid genes, *Cab1R* and *Cab2R* (encoding light harvesting Chl a/b binding protein of PSII), *psaA* and *psbA* (encoding polypeptides of two different photosystems: PSI and PSII) were significantly suppressed in the *wpb1* and white panicled F₂ plants (Fig 4C).

Overall, the expression of Chl biosynthesis genes (*HEMA1*, *CAO1*, *PORA*), chloroplast development-associated genes (*V1*, *V2*, *rpoA*, *rpoB*) and photosynthesis genes (*Cab1R*, *Cab2R*, *psaA*, *psbA*) in *wpb1* were significantly different from those in NIP. Both Chl biosynthesis and photosynthesis genes were significantly downregulated or severely inhibited. In addition, the gene *rpoB* for chloroplast development, was significantly upregulated, while *V1* and *rpoA* were downregulated in *wpb1* plants. These results indicated that the processes of chloroplast development, chlorophyll synthesis, and photosynthesis in branches of *wpb1* were severely altered by the mutation of *wpb1*.

Although the expression patterns of the related genes were not completely consistent between F₂ plants and their parents, they exhibited the similar trend of expression alterations between F₂ plants and their parents except *rpoA*, *V1*, and *V2* genes. For example, the differences in the expression of photosynthesis-associated genes between white panicle branches and green panicle branches in F₂ plants was in agreement well with the case between *wpb1* and NIP.

Therefore, we infer that the wpb1 gene may involve in the regulation pathway of the chlorophyll biosynthesis, chloroplast development, and photosynthesis.

### Parenchymatous tissue cells of the white panicle branch have low chlorophyll contents in *wpb1* plants

To verify our hypothesis, fluorescence microscopy was used to examine the chlorophyll fluorescence emitted from the panicle branches. In the leaves of NIP and *wpb1* plant, the comparable levels of florescence were detected using fluorescence microscopy (S3 Fig). The white panicle branches of *wpb1* and whitened branch individuals selected from the F₂ population showed no or weak chlorophyll fluorescence emission (Fig 5C, 5D, 5G and 5H) in contrast to those in the NIP and green branch F₂ plants (Fig 5A, 5B, 5E and 5F). However, the green branches of NIP and F₂ plants showed bright fluorescence emissions.

The results suggested that the chlorophyll contents in white panicle branches of *wpb1* are rather low by comparing with those of NIP. The chlorophyll content measurement agreed

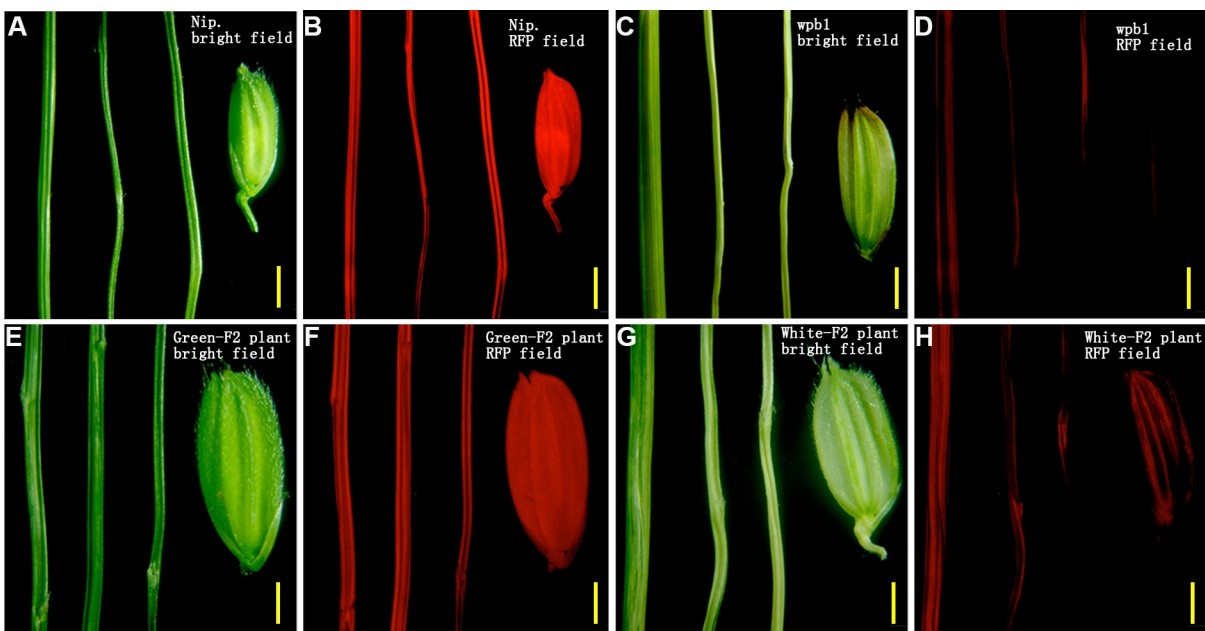

**Fig 5. Chlorophyll fluorescence examination of the rachis, branch, and spikelet by stereomicroscopy.** A-B. NIP in bright field (A) and RFP field (B). C-D. *wpb1* mutant in bright field (C) and RFP field (D). E-F. $F_2$-green in bright field (E) and RFP field (F). G-H. $F_2$-white in bright field (G) and RFP field (H). Bright field: Exposure time 300 ms, ISO 200. RFP field: Exposure time 1 s, ISO 1600. Bar, 2 mm (A-D). 1.5 mm (E-H).

with the results. The contents of chlorophyll a, chlorophyll b and total chlorophyll in the rachis of *wpb1* were significantly lower than those in NIP (only 29.06%, 11.96%, and 25.33% of NIP, respectively). Similarly, they were only 0.0134 mg/g.FW, 0.0019 mg/g.FW, and 0.0153 mg/g.FW, respectively, in white panicle branches of *wpb1*, which were only 11.2%, 6.5% and 10.3% of those in NIP (green branch) (Fig 6A and 6B). Moreover, the chlorophyll contents in white and green panicle branches of $F_2$ plants were similar to those in their parents. The chlorophyll a, chlorophyll b, and total chlorophyll contents in the samples of white branched $F_2$ plants were extremely low, which were close to or lower than those of white branched *wpb1*. The chlorophyll contents in the samples of green branch $F_2$ plants were close to or slightly lower than those of the parent NIP (Fig 6B). However, intriguingly, the chlorophyll contents of the rachises of white branched plants was only slightly lower than those in green branched plants (Fig 6A).

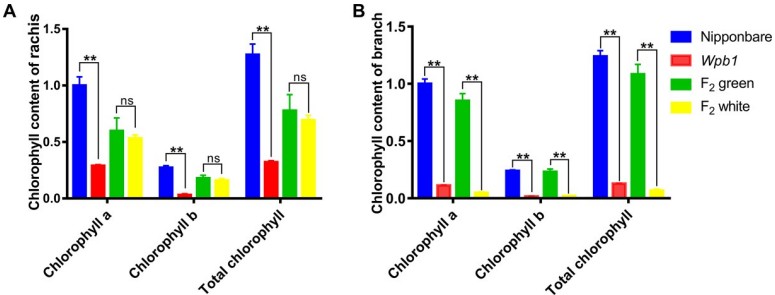

**Fig 6. Measurement of the chlorophyll contents in parents and $F_2$ populations.** A. In rachis. B. In branch. All data are normalized by chlorophyll a of NIP, and its chlorophyll content in NIP was set to 1.0. * ($P<0.05$), ** ($P<0.001$).

The lower chlorophyll content in white panicle branches implied that chlorophyll synthesis was defective in parenchymatous tissue of *wpb1* panicle branches. The section analysis showed that, compared with parenchymatous cells in NIP panicle branches where chloroplasts were well developed with clear green colour, rare or no green cells were observed in the white panicle branches of *wpb1* (Fig 7A–7D).

The results further supported the notion that the normal processes of chloroplast development and chlorophyll formation in parenchymatous tissues of white panicle branches of *wpb1* is disturbed.

## The chloroplast biogenesis is inhibited in the panicle branch of *wpb1*

To further confirm the failure in the chloroplast biogenesis in the panicle branches of *wpb1*, an ultra-thin sectioning of parenchymatous tissues where chloroplasts developed in the panicle branches were performed. In the white panicle branch of *wpb1* plants, no chloroplasts were

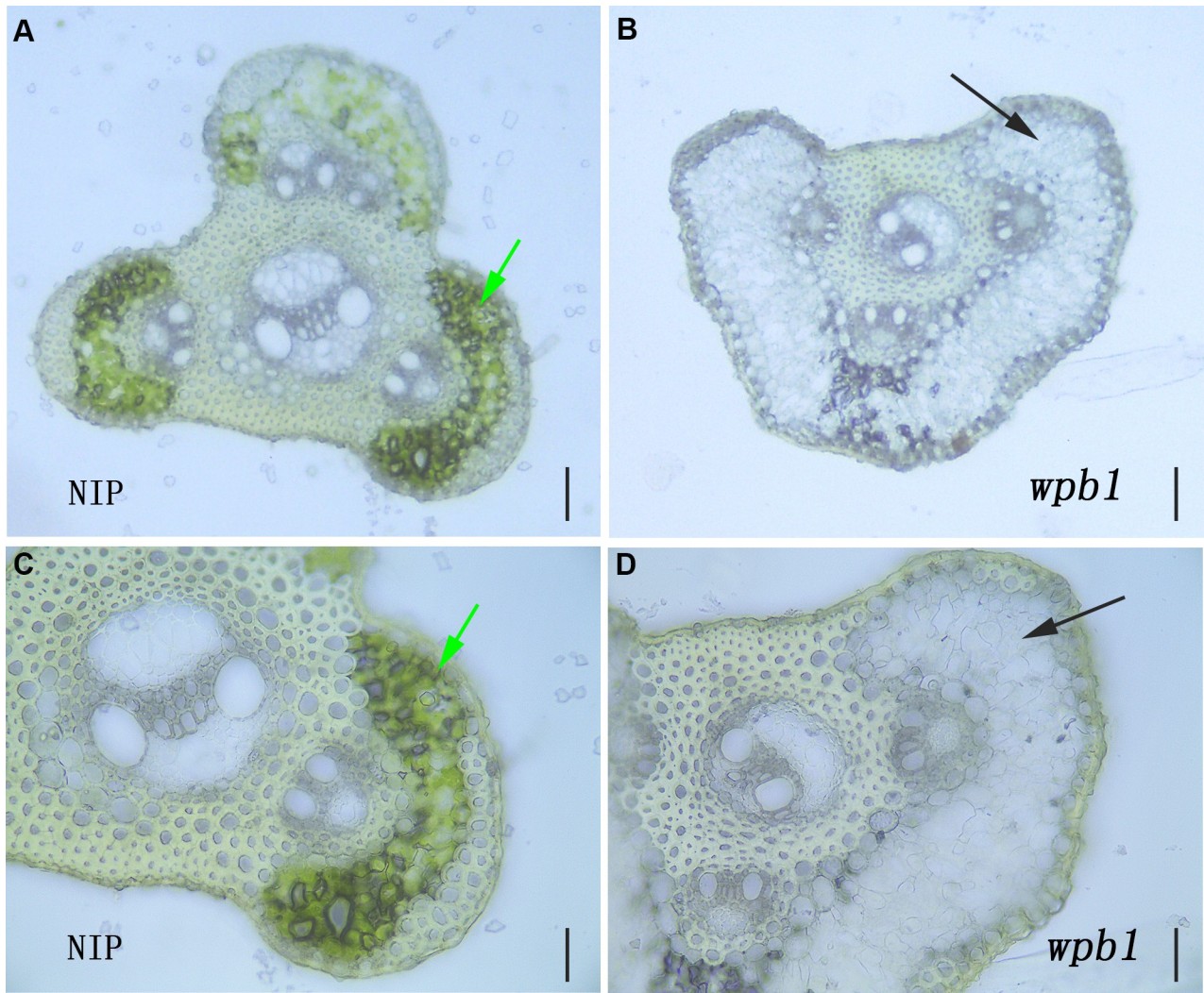

**Fig 7. Section of panicle branches of NIP and *wpb1*.** A, C. The cross-sectioned panicle branch of NIP. B, D. The cross-sectioned panicle branch of *wpb1* mutant. The arrows indicate that the obvious distinction between the parenchymatous tissue of NIP and *wpb1*. Bar = 100 μm (A-B), 50 μm (C-D).

observed in parenchyma cells (Fig 8), and the chloroplast biogenesis seems completely being blocked. We suggested that the failure in chloroplast biogenesis in parenchymatous cells of panicle branches resulted in the albino phenotype of *wpb1*.

## Discussion

Photosynthesis is an important biochemical reaction in higher plants. Normal chloroplast biogenesis in leaves of plants is of extremely important for photosynthesis. The defect in chloroplast development leads to tissue chlorosis or albinism. Thus far, many rice albino mutants

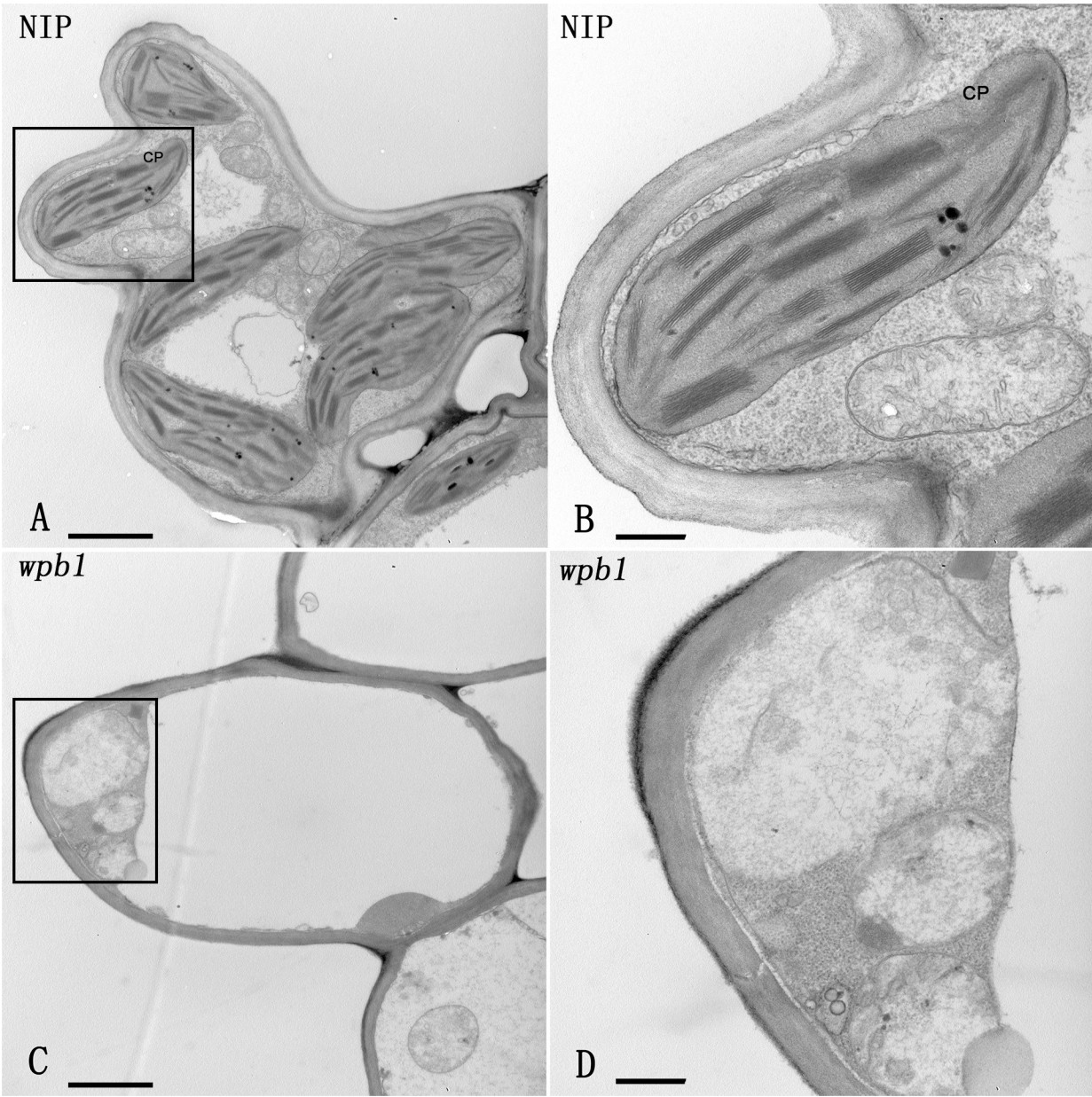

**Fig 8. The comparison of the chloroplast development in parenchymatous cells of panicle branches of Nipponbare and *wpb1* using transmission electron microscopy.** A. The images of Nipponbare. B. The magnification of the boxed region in A. C. The image of *wpb1*. D. The magnification of the boxed region in C. Bar = 2 μm (A, C), 500 nm (B, D). CP designates Chloroplast.

have been reported in previous studies, mainly including leaf albino, seedling albino and panicle albino mutants [8, 47–53]. Mutations in the *al4* [54], *al8*, *as11* [55], and *asl3* [56] genes lead to leaf and seedling albinism. *wlp1* [29], *wlp2* [51], and *wp3* [57] mutants exhibited panicle and glume albinism. Most of these mutants are unable to complete their life cycle, although a few of them can grow up under a specific environment. Most of these albino mutations result from defects in chloroplast development [19, 29, 41, 49, 51, 57], which is the same as *wpb1*.

In this study, a natural mutant with albino branches was found from a tropical japonica rice variety *wpb1*. The key feature of the phenotype that appeared in *wpb1* was white panicle branches without chloroplasts, and no additional abnormal coloration was observed in other parts of the rice plants. Genetic analysis showed that *wpb1* was controlled by a single recessive nuclear gene. Whole genome resequencing and BSA analysis were combined to preliminarily map the candidate interval for *wpb1*. Two intervals located on chromosome 1 and chromosome 3 were targeted as *wpb1* candidates. Further, the one on chromosome 3 was excluded from the 40k SNP array analysis. Subsequently, by map-based cloning, *wpb1* was narrowed down to a 95-kb interval flanked by M8 and M10 on chromosome 1 with 17 predicted genes in this region.

Several non-allelic genes with similar phenotypes have been cloned. The cloned *wp1* gene, which is located on chromosome 7 and encodes a Val-tRNA synthetase (OsValRS2) with a single base changed, leads to albino phenotypes in seedlings and white panicles at heading stage. [23]. This mutant appears to have green leaves and glumes, green branches and spike axes. The development of seedlings with severe albinism was arrested and seedlings died at the 4-leaf stage. All observations above showed that there was an obvious difference in the branch colour phenotype between the *wpb1* and *wp1* mutants. However, the responses of the albino phenotypes to temperature change in the two mutants were similar. The two mutants showed aggravated albinism and a widened range of albino parts under low temperature. Another temperature-responsive albino mutant, *wlp2*, exhibited light green leaves and panicles in rice when the temperature was below 22˚C. With increasing temperature (22–32˚C), the albino phenotype appeared and aggravated significantly, resulting in the death of the rice plants [51]. The differences in albino phenotype between *wlp2* mutant and *wpb1* were also identified, such as the albino tissue, degree, and temperature response.

Li et al. cloned the *WP3* that was related to the albino panicle [57]. The location of *wp3* gene is close to the *wpb1* interval on chromosome 1. *wpb1* is located at 14-Kb upstream of *WP3* (Os01g0306650). Both mutants showed albino panicles and albino streaked leaves; however, the differences between two genotypes were obvious. First, the *wp3* mutant showed uniform albinism for the whole panicle and milky albinism (light green) throughout the panicle, with albino seedling. Furthermore, the variations are not observed in the coding sequences of the allele of *WP3* gene between Nipponbare and *wpb1* mutant (S4 Fig). The *wpb1* mutant in this study showed branch albinism (complete albino), while albinism in the panicle rachis and glume shell were regulated by temperature. Albinism in the *wpb1* mutant could extend from the panicle branch to the panicle rachis and glume shell with the decreasing of temperature (18–25˚C). Second, no whitening phenotypes were observed in *wpb1* at the seedling stage. Finally, the albinism of streaked leaves in the *wp3* mutant shows maternal inheritance. The phenotype is not co-segregated with branch albino in *wp3* plants. Whereas, *wpb1* is a pleiotropic recessive nuclear gene with multiple effects for controlling the phenotypes of albino streaked leaf and albino panicles in *wpb1* mutant. Therefore, we inferred that the *wpb1* gene was a novel gene resulting in the albinism of rice panicle branches and white streaked leaves in rice.

The expression analysis showed the altered expressions of the genes involved in chlorophyll biosynthesis, chloroplast development, and the photosynthesis system, suggesting that the mutation of *WPB1* might directly or indirectly impair the processes of chloroplast development and/or chlorophyll formation, and thereby resulting in the albino panicle branch

phenotype of the variety. The expressions of chlorophyll synthesis genes in *wpb1* were significantly downregulated, while their expressions in the white and green branches of $F_2$ plants were not extremely induced. We speculated that the complicated recombinant events that occurred in $F_2$ plants led to the discrepancy in the expression of chlorophyll biosynthesis-related genes between parental lines and their offspring. However, these discrepancies were unable to significantly affect the appearance of the branch albino phenotype in offspring with a homozygous recessive *wpb1* gene. At the same time, the fact that the white streak leaves observed in the (NIP×*wpb1*) $F_2$ generation only appeared in some but not all of the white panicle branched plants is noteworthy; in contrast, the white streak leaf phenotype and white branches were completely co-segregated in the $BC_1F_2$ population. We proposed that the white panicle branch and white streaked leaves were controlled by the same gene in the $BC_1F_2$ backcross population. Additional genes with synergistic/inhibited interaction that can reduce/eliminate or aggravate white streaks in leaves to a certain extent in $F_2$ plants were lost in selecting backcrosses. Therefore, *wpb1* deserves more focus in our further studies.

In spite of many research works have been done on the biogenesis of chloroplasts, the mechanism of chloroplast development remains elusive [58]. Albino mutants provide excellent materials for exploring the mechanism underlying the development of chloroplast. Therefore, the study of the mutant *wpb1* is of great significance. Further cloning and in-depth functional analysis of the *wpb1* gene will help to understand the regulatory mechanism of chloroplast development in rice branches. For the application of this genetic material, the underlying gene *wpb1* can be transformed into two-line male sterile lines as a colour marker for field trial investigation, which is used to prevent and eliminate the confusion to ensure hybrid purity in the breeding of two-line materials and in the seed production of hybrid rice.

## Supporting information

**S1 Fig. The comparison of the panicle branch coloration phenotypes of Nipponbare, *wpb1*, and their offspring.** A. The panicle branches of Nipponbare and *wpb1*. B. The panicle branches of $F_1$ plant. C and D. The panicle branches of $F_2$ plant. Bar = 1 cm.
(TIF)

**S2 Fig. Phenotypic characterization of the *wpb1* mutant.** A. *wpb1* mutant in the tillering stage. B. The tiller of NIP (top) and *wpb1* mutant (bottom). C. The spike phenotype of the *wpb1* mutant at different heading temperatures. 1 NIP, 2 heading at 30˚C, 3 heading at 25˚C, 4 and 5 heading at 18˚C. As the temperature decreases, the branches change from white to red. D. Branches phenotype. 1 NIP, 2 heading at 30˚C, 3 heading at 25˚C, 4 heading at 18˚C. Bar, 3 cm (B). 5 cm (C). 8 cm (D).
(TIF)

**S3 Fig. The chlorophyll fluorescence of the leaf by stereomicroscope.** A. bright field. B. RFP field. Chlorophyll can fluoresce in the RFP field. Bright field: Exposure time 300 ms, ISO 200. RFP field: Exposure time 1 s, ISO 1600.
(TIF)

**S4 Fig. The comparison of the coding sequencing of two alleles of *WP3* gene in Nipponbare and *wpb1*.**
(TIF)

**S1 Table. List of fine mapping primers used in this study.** The list of primers of InDel markers for fine mapping obtained from Illumina sequencing.
(DOCX)

**S2 Table. List of qPCR primers used in this study.** The list of specific primers for qPCR to examine the expression differences of the genes involved in Chl biosynthesis, photosynthesis and chloroplast development in the wpb1 mutant and NIP.
(DOCX)

**S3 Table. Predicted genes according to BSA-Seq and ANNOVAR annotations.** The list of fourteen candidate genes according to BSA-Seq and ANNOVAR annotations.
(DOCX)

**S4 Table. The annotations of the candidate genes in the *wpb1* mapping interval.** The list and annotations of eleven candidate genes.
(DOCX)

# Acknowledgments

The authors want to thank D. Jianglei Rao for assistance in this study, and D. Yourong Fan for assistance in manuscript preparation.

# Author Contributions

**Conceptualization:** Zhongquan Cai, Jijing Luo.

**Data curation:** Zhongquan Cai, Peilong Jia, Jiaqiang Zhang, Ping Gan, Qi Shao, Jijing Luo.

**Formal analysis:** Jijing Luo.

**Funding acquisition:** Jijing Luo.

**Methodology:** Jijing Luo.

**Visualization:** Gang Jin, Jijing Luo.

**Writing – original draft:** Zhongquan Cai, Peilong Jia, Gang Jin, Liping Wang, Jijing Luo.

**Writing – review & editing:** Jian Jin, Jiangyi Yang, Jijing Luo.

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
