## [Decision Letter · Decision Letter 0]

5 Aug 2019

PONE-D-19-16776

Genetic analysis and fine mapping of a qualitative trait locus wpb1 for albino panicle branches in rice

PLOS ONE

Dear Dr. Luo,

Thank you for submitting your manuscript to PLOS ONE. After careful consideration, we feel that it has merit but does not fully meet PLOS ONE’s publication criteria as it currently stands. Therefore, we invite you to submit a revised version of the manuscript that addresses the points raised during the review process.

We would appreciate receiving your revised manuscript by Sep 19 2019 11:59PM. To enhance the reproducibility of your results, we recommend that if applicable you deposit your laboratory protocols in protocols.io, where a protocol can be assigned its own identifier (DOI) such that it can be cited independently in the future. For instructions see: http://journals.plos.org/plosone/s/submission-guidelines#loc-laboratory-protocols

We look forward to receiving your revised manuscript.

Kind regards,

Maoteng Li

Academic Editor

PLOS ONE

Journal Requirements:

2. Please include your tables as part of your main manuscript and remove the individual files. ** Please note that supplementary tables (should remain/ be uploaded) as separate "supporting information" files **

Reviewers' comments:

Reviewer's Responses to Questions

**Comments to the Author**

1. Is the manuscript technically sound, and do the data support the conclusions?

Reviewer #1: Yes

Reviewer #2: Yes

2. Has the statistical analysis been performed appropriately and rigorously? 

Reviewer #1: Yes

Reviewer #2: Yes

3. Have the authors made all data underlying the findings in their manuscript fully available?

Reviewer #1: Yes

Reviewer #2: Yes

4. Is the manuscript presented in an intelligible fashion and written in standard English?

Reviewer #1: Yes

Reviewer #2: Yes

5. Review Comments to the Author

Reviewer #1: The linkage map of wpg1 gene was constructed and it was demonstrated that the 17 genes were predicted within candidate gene region on the chromosome. It was inferred the cause gene corresponding to wpg1 mutation functions the chloroplast development by the expression analysis of some genes and microscopy analyses. These analyses and the discussion can be appreciated.

However, this reviewer has the questions about the following points.

-P13, line 14-16

It is described as “The results suggested that the mutation of WPB1 severely impairs chloroplast development, chlorophyll biosynthesis, and photosynthesis in the panicle branches of wpb1”. The reason why “the wpb1 gene may involve in the regulation pathway of the chloroplast development” only by the results of enzyme activity should be explained.

-Annotation list of the predicted gene

Annotation list of the predicted genes shown in Fig3C and TableS3 should be shown. After that, the function of the candidate genes regulating the chloroplast development should be explained. Even when it is difficult to discuss by only annotation of gene, the authors should refer to the effect.

Reviewer #2: Comments:

1. P4, Ln3-4:the sentence "Bulked segregant analysis (BSA) is a rapid method for locating target trait genes in lettuce" is not very suitable, it can be modified to "Bulked segregant analysis (BSA) is a rapid method for locating target trait genes and it was initially used in lettuce".

2.P16, Ln12-22, the authors concluded wpb1 and wp3 not the same locates due to their different genotype. however, wpb1 is very nearby to wp3 (14 kb). So it had better to detect whether the sequence of wp3(Os01g0306650) are different in wbp1 mutant and nipponbare, and the candidate recombiant lines.

3. P16, ln23-24: While the albino streaked leaf of wpb1 is controlled by the same recessive nuclear gene. the sentence is obscure. it had better modifed to "Whereas the phenotypes of albino streaked leaf and albino panicles of wpb1 are controlled by the same recessive nuclear gene. on the other hand, the authors cannot conclude albino streaked leaf and albino panicles of wpb1 is controled by the same gene at present. whether wpb1 is one gene with multiple effects?

6. PLOS authors have the option to publish the peer review history of their article (what does this mean?). If published, this will include your full peer review and any attached files.

Reviewer #1: No

Reviewer #2: No

---

## [Author Response · Author response to Decision Letter 0]

10 Sep 2019

Dear Editor,

Hope you are doing well!

We greatly appreciate for your kindly and helpful suggestions to our manuscript entitled ‘Genetic analysis and fine mapping of a qualitative trait locus wpb1 for albino panicle branches in rice’. We carefully revised the manuscript according to the comments, and now resubmit revised manuscript for your consideration. In the revised version, all detailed revisions were described in ‘Point-by-point response to reviewers’ shown below for the convenience of you viewing.

We think the current version of this manuscript would have been significantly improved after revision. 

Thanks again and any further suggestions to our manuscript from you will be highly appreciated and we are very grateful for your kind consideration for publication.

Best wishes

Yours sincerely,

Jijing Luo, Professor (Ph.D.)

College of Life Science and Biotechnology, Guangxi University

State Key Laboratory for Conservation and Utilization of Subtropical Agro-bioresources, Guangxi University

100 Daxue Rd. (East), Nanning, 530004, China

Cell phone: +86-18077792389

E-mail: jjluo@gxu.edu.cn

Point-by-point response to reviewers

We thank the Editor and Reviewers for insightful comments, which help us to substantially revise the manuscript.

In the current version of manuscript, we made some revisions according to the comments, including English language editing, supplemented Tables, and adjusted the description of the related content for more readability.

1. For the S3 Table, according to the comment of Reviewer 1, we added annotation information for the genes. Added S4 Table to list the annotation information of genes in wpb1 interval.

2. The genetic information between wpb1 and WP3 was added to clarify their relationship more clearly in the discussion according to the comment of Reviewer 2. P16, line 3.

3. Reviewers’ suggestions for several descriptions were adopted for more readability.

Notes: Based on the guideline of editorial office, all the revisions were highlighted with red color for any Changes except the deletions in the revised manuscript. All the detailed revisions to the manuscript were described in this section.

4. We added sentence “Furthermore, the variations are not observed in the coding sequences of the allele of WP3 gene between Nipponbare and wpb1 mutant (S4 Fig).” to the discussion section to describe no sequence difference was observed in the coding sequences of wp3 gene in two parents, Nipponbare and wpb1 and supplemented with a supplementary figure, S4 Fig (P16, L14-16). Figure legend was supplemented at P22, L36-37. 

Reviewer #1: The linkage map of wpg1 gene was constructed and it was demonstrated that the 17 genes were predicted within candidate gene region on the chromosome. It was inferred the cause gene corresponding to wpg1 mutation functions the chloroplast development by the expression analysis of some genes and microscopy analyses. These analyses and the discussion can be appreciated.

However, this reviewer has the questions about the following points.

-P13, line 14-16

It is described as “The results suggested that the mutation of WPB1 severely impairs chloroplast development, chlorophyll biosynthesis, and photosynthesis in the panicle branches of wpb1”. The reason why “the wpb1 gene may involve in the regulation pathway of the chloroplast development” only by the results of enzyme activity should be explained.

Reply: Thank you for your careful reading and great comment! As you pointed out, it is inappropriate to infer that “the wpb1 gene may involve in the regulation pathway of the chloroplast development” only by the results of enzyme activity. The results of this experiment show that all three biological processes are abnormal at transcriptional level, and it is not clear which process is directly affected. So, we revised as a concise sentence: ‘Therefore, we infer that the wpb1 gene may involve in the regulation pathway of the chlorophyll biosynthesis, chloroplast development, and photosynthesis.’ (P13, line5-6, revised manuscript)

-Annotation list of the predicted gene

Annotation list of the predicted genes shown in Fig3C and S3 Table should be shown. After that, the function of the candidate genes regulating the chloroplast development should be explained. Even when it is difficult to discuss by only annotation of gene, the authors should refer to the effect.

Reply: According to your suggestions, we have strengthened the analysis in this field. But, no annotated information and literature about these genes involved in albinism, chloroplast development, chlorophyll synthesis and photosynthesis were obtained. So, we did not predict specific genes, only added annotation information in S3 Table, and added S4 Table to list annotation information of genes in wpb1 interval. The citation in the text was made in the corresponding places for S4 Table (P11, line16, revised manuscript).

Reviewer #2: Comments:

1. P4, Ln3-4:the sentence "Bulked segregant analysis (BSA) is a rapid method for locating target trait genes in lettuce" is not very suitable, it can be modified to "Bulked segregant analysis (BSA) is a rapid method for locating target trait genes and it was initially used in lettuce".

Reply: It was significantly improved after your revision for the sentence. Thank you! 

(P4, line2-3, revised manuscript)

2.P16, Ln12-22, the authors concluded wpb1 and wp3 not the same locates due to their different genotype. however, wpb1 is very nearby to wp3 (14 kb). So it had better to detect whether the sequence of wp3(Os01g0306650) are different in wbp1 mutant and nipponbare, and the candidate recombiant lines. 

Reply: According to your suggestion, we sequenced the CDS sequences of WP3 and did a sequence analysis between wpb1 mutant and nipponbare. The variations were not observed in the two coding sequences of Wp3 alleles in Nipponbare and wpb1 mutant (S4 Fig).

3. P16, ln23-24: While the albino streaked leaf of wpb1 is controlled by the same recessive nuclear gene. the sentence is obscure. it had better modifed to "Whereas the phenotypes of albino streaked leaf and albino panicles of wpb1 are controlled by the same recessive nuclear gene. on the other hand, the authors cannot conclude albino streaked leaf and albino panicles of wpb1 is controled by the same gene at present. whether wpb1 is one gene with multiple effects?

Reply: Thank you for your careful reading and excellent revision. Yes, the wpb1 is one gene with multiple effects, white panicle branch and albino streaked leaf.

(P16, line20-21, revised manuscript)

---

## [Decision Letter · Decision Letter 1]

17 Sep 2019

Genetic analysis and fine mapping of a qualitative trait locus wpb1 for albino panicle branches in rice

PONE-D-19-16776R1

Dear Dr. Luo,

We are pleased to inform you that your manuscript has been judged scientifically suitable for publication and will be formally accepted for publication once it complies with all outstanding technical requirements.

With kind regards,

Maoteng Li

Academic Editor

PLOS ONE

Additional Editor Comments (optional):

Reviewers' comments:

Reviewer's Responses to Questions

**Comments to the Author**

1. If the authors have adequately addressed your comments raised in a previous round of review and you feel that this manuscript is now acceptable for publication, you may indicate that here to bypass the “Comments to the Author” section, enter your conflict of interest statement in the “Confidential to Editor” section, and submit your "Accept" recommendation.

Reviewer #1: All comments have been addressed

Reviewer #2: All comments have been addressed

2. Is the manuscript technically sound, and do the data support the conclusions?

Reviewer #1: Yes

Reviewer #2: Yes

3. Has the statistical analysis been performed appropriately and rigorously? 

Reviewer #1: Yes

Reviewer #2: Yes

4. Have the authors made all data underlying the findings in their manuscript fully available?

Reviewer #1: Yes

Reviewer #2: Yes

5. Is the manuscript presented in an intelligible fashion and written in standard English?

Reviewer #1: Yes

Reviewer #2: Yes

6. Review Comments to the Author

Reviewer #1: The manuscript has been revised adequately according to reviewer's comments. I didn't have an additional comment.

Reviewer #2: (No Response)

7. PLOS authors have the option to publish the peer review history of their article (what does this mean?). If published, this will include your full peer review and any attached files.

Reviewer #1: No

Reviewer #2: Yes: Aihua Sha

---

## [Editor Report · Acceptance letter]

19 Sep 2019

PONE-D-19-16776R1 

Genetic analysis and fine mapping of a qualitative trait locus wpb1 for albino panicle branches in rice 

Dear Dr. Luo:

I am pleased to inform you that your manuscript has been deemed suitable for publication in PLOS ONE. Congratulations! Your manuscript is now with our production department. 

With kind regards,

on behalf of

Dr. Maoteng Li 

Academic Editor

PLOS ONE